# Poverty–Food Insecurity Nexus in the Post-Construction Context of a Large Hydropower Dam in the Brazilian Amazon

**DOI:** 10.3390/ijerph21020155

**Published:** 2024-01-30

**Authors:** Igor Cavallini Johansen, Miquéias Freitas Calvi, Verônica Gronau Luz, Ana Maria Segall-Corrêa, Caroline C. Arantes, Victoria Judith Isaac, Renata Utsunomiya, Vanessa Cristine e Souza Reis, Emilio F. Moran

**Affiliations:** 1Center for Environmental Studies and Research (NEPAM), State University of Campinas (UNICAMP), Campinas 13083-867, SP, Brazil; v.esouzareis@griffith.edu.au (V.C.e.S.R.); moranef@msu.edu (E.F.M.); 2Faculty of Forestry, Federal University of Pará (UFPA), Altamira 68372-040, PA, Brazil; mcalvi@ufpa.br; 3Faculty of Health Sciences, Grande Dourados Federal University (UFGD), Dourados 79825-070, MS, Brazil; veronicaluz@ufgd.edu.br; 4Fundação Oswaldo Cruz, Brasília 70904-130, DF, Brazil; ana.correa@fiocruz.br; 5Division of Forestry and Natural Resources, West Virginia University, Morgantown, WV 26506-6125, USA; caroline.arantes@mail.wvu.edu; 6Núcleo de Ecologia Aquática e Pesca, Federal University of Pará, Belém 66077-530, PA, Brazil; visaac@ufpa.br; 7Institute of Energy and Environment (IEE), University of São Paulo (USP), São Paulo 05508-010, SP, Brazil; renata.utsunomiya@usp.br; 8Australian Rivers Institute, Griffith University, Nathan, Brisbane, QLD 4111, Australia; 9Department of Geography, Michigan State University, East Lansing, MI 48823, USA

**Keywords:** public health, sustainable development, hydropower dams, food insecurity, poverty and resettlement, Brazilian Amazon

## Abstract

Within the 2030 Sustainable Development Agenda, large hydropower dams are positioned as a sustainable energy source, notwithstanding their adverse impacts on societies and ecosystems. This study contributed to ongoing discussions about the persistence of critical social issues, even after the investments of large amounts of resources in areas impacted by the construction of large hydropower dams. Our study focused on food insecurity and evaluated this issue in the city of Altamira in the Brazilian Amazon, which has been profoundly socially and economically impacted by the construction, between 2011 and 2015, of Brazil’s second-largest dam, namely, Belo Monte. A survey in Altamira city featured a 500-household random sample. Structural equation modeling showed conditioning factors of 60% of the population experiencing varying food insecurity degrees. Poverty, female-led households, lower education, youth, and unemployment were strongly linked to higher food insecurity. Crowded, officially impacted, and resettled households also faced heightened food insecurity. Our findings underscore the food insecurity conditions in the region impacted by the Belo Monte dam, emphasizing the need to take into account this crucial issue while planning and implementing hydropower dams.

## 1. Introduction

The United Nations’ 2030 Agenda for Sustainable Development promotes the goal of “ensuring access to affordable, reliable, sustainable, and modern energy for all” (SDG 7) [1,2]. Another objective of the UN agenda relates to food security, namely, SDG2: “end hunger, achieve food security and improved nutrition and promote sustainable agriculture”. To effectively achieve these goals on local to global scales, it is crucial to have a comprehensive understanding of what constitutes sustainable energy and what does not. Contrary to common practice that often categorizes hydropower as sustainable, substantial evidence has revealed the negative social and environmental issues that remained or were increased in the context of the expansion of this energy source, including public health issues, such as food insecurity [3,4,5,6,7,8,9,10,11,12,13]. 

The classic literature on “energy boomtowns” provides valuable insights into the consequences of energy projects on the well-being and quality of life of host communities. This literature elucidates how energy infrastructure projects frequently face challenges in adapting to the abrupt influx of predominantly male workers during a resource boom due to the limited local capacity for such adjustments [14,15,16]. The surge of workers poses numerous challenges for the community, including the strain on local infrastructure, such as housing, sewage systems, and garbage services [17,18,19,20]. Moreover, the preferential access of these workers to this infrastructure can displace access for the local population, exacerbating the situation [21]. The competition over limited resources can restrict access to food during the construction of energy projects, as documented in previous studies [22,23,24]. This is also compounded by the impact on local food production, as the rural workforce moves to the construction site in search of better wages [25].

In this context, the present study contributed to the existing body of literature by examining the levels of food insecurity found even after the investments of large amounts of resources in an area in the Amazon region directly affected by the construction of the second-largest hydropower dam in Brazil, namely, the Belo Monte dam, which was built between 2011 and 2015. This study aimed to evaluate the food insecurity situation in the city of Altamira, which is located in the Brazilian Amazon and has been profoundly socially and economically impacted by the construction of this dam. In addition, this study investigated potential factors associated with the food insecurity levels in the study location. To the best of our knowledge, this study represents the first analysis of food insecurity status after the construction of a large hydropower dam in the Brazilian Amazon employing a scientifically validated scale.

Our initial hypothesis was that food insecurity was influenced by a range of underlying factors. We supposed that these conditioning factors created a layered effect that ultimately resulted in food insecurity. Our second hypothesis was that poverty played a pivotal role among the underlying factors of food insecurity, and the most affected groups were those who underwent resettlement. Our final hypothesis stated that in the post-dam construction, COVID-19 increased the challenges related to food insecurity in the study area. 

Due to its significant hydropower potential, the Amazon region has become a focal point for new hydropower projects. Over the past few decades, the Amazon basin has been recognized as a crucial area for expanding electricity production in Brazil [26,27,28]. 

The country asserts that it has a “clean” energy matrix, with approximately 62% of its electricity generated from hydropower sources [29]. This statement promotes the notion that hydropower dams represent the most viable renewable energy solution, which is characterized as clean, sustainable, and affordable. However, this argument obscures the severity of the adverse socio-environmental issues that occur during and remain after construction [6,30].

Among these issues, previous studies revealed that hydropower dams constructed in the Amazon region have resulted in increased stress among local populations [31]. Furthermore, they have been associated with the erosion of social capital [32] and a decrease in self-rated health [33]. Through resettlement, these dams were found to undermine the cultural, social, and economic reproduction of traditional peoples and communities [34,35]. Additionally, they contribute to gender imbalances [36,37] and impact the food security of local populations [30].

There are a variety of food security definitions that have evolved. This study utilized the notion of food and nutritional security used in Brazil by the Organic Law of Food and Nutritional Security (LOSAN), which “consists of ensuring everyone’s right to regular and permanent access to quality food, in sufficient quantities, without compromising access to other essential needs. This is based on health-promoting dietary practices that respect cultural diversity and are environmentally, culturally, economically, and socially sustainable” [38]. Food insecurity and nutrition have been assessed in the country since 2004, following the validation of the Brazilian Household Food Insecurity Measurement Scale (EBIA) [39]. 

The EBIA has found extensive application in scientific research and official data collection, such as the Brazilian National Household Sampling Survey (PNAD) conducted by the Brazilian Institute of Geography and Statistics (IBGE). The PNAD took place in 2004, 2009, and 2013, and was carried out by the IBGE. The EBIA was also utilized to inform the Family Budget Surveys (POF) in 2018 and, more recently, in 2021 and 2022, it was conducted by the Brazilian Research Network on Food and Nutrition Sovereignty and Security (Rede Penssan). The latter revealed that 125.2 million people in the country experienced some degree of food insecurity, with over 33.1 million identified as being in a state of hunger, representing a 74.2% increase compared with two years prior [40]. Recent evidence demonstrated a correlation between the COVID-19 pandemic and the escalation of food insecurity in Brazil [41], even though the prevalence of food insecurity and hunger began to rise before the pandemic and as a result of the political crisis that began in 2016 [40].

## 2. Materials and Methods

### 2.1. Study Site

The municipality of Altamira is located in the center-south of Pará state (3°12′24.8″ S 52°13′07.5″ W) (Figure 1). The municipality has 126,279 inhabitants according to the most recent national demographic census [42]. Our study encompassed the urban zone, where over 84% of the municipality’s population resided as of 2010 [43]. Altamira bore the brunt of the Belo Monte hydropower dam’s construction impacts due to its status as the most populous urban center in the region. This positioning rendered Altamira a central hub for the dispersion of goods, services, and logistical support integral to the dam’s erection. As a result, Altamira’s population was strongly impacted by the dam construction.

As the second-largest hydropower dam in Brazil, Belo Monte boasts an installed capacity of 11.2 GW, generating an average of 4.5 GW. Its structure is equipped with 24 turbines and 18 spillway gates. Designed as a run-of-river hydropower system, it aims to minimize the size of its water reservoir. Nevertheless, the project’s flooded area spans a substantial 478 square kilometers [44]. The construction had significant social-ecological impacts, notably a substantial reduction in water flow in a specific stretch of about 100 km of the river known as Volta Grande do Xingu. In this area, the river flow decreased by approximately 80% due to the redirection of the natural water flow to the diversion channel leading to the main powerhouse [34,45,46].

### 2.2. Data

This survey served as an integral component of a broader, long-term project involving multiple researchers who utilize a diverse range of quantitative and qualitative methodologies to examine the enduring social and environmental consequences arising from the construction of large hydropower dams in the Brazilian Amazon a decade after their completion. In this specific phase of the study, we employed a robust probability sampling method [47] based on a random sample of 500 households selected to represent different socioeconomic strata and geographic areas within the urban study site. The sampling process involved initially selecting 10 urban census tracts and then randomly selecting 50 households within each census tract. All households had an equal chance of being included in the sample. Interviews were conducted with the head of the household or another household member aged 18 or over. Eligibility for participation in the survey required residency in the urban area of Altamira during and after the dam construction. Data collection occurred between 13 July and 30 July 2022 and was carried out by eight interviewers who were undergraduate or graduate students from local universities. Prior to the household visits, interviewers underwent intensive training.

The survey consisted of administering the 8-question Brazilian Household Food Insecurity Measurement Scale (EBIA) [48] to all 500 households in the sample. Additionally, two supplementary questions were included to assess the extent to which observed food insecurity was associated or not with the post-dam construction period (after 2016). Notably, the construction of the Belo Monte dam commenced in 2011 and it began operating at the end of 2015. Although some parts of the construction continued until 2019, we considered the primary phase of construction, namely, when most of the working force was in place, as being from 2011 to 2015. Thus, for our study, 2016 serves as the reference point for the beginning of the post-construction period.

Furthermore, the questionnaire included sociodemographic inquiries about household members, the head of the household, and household assets. Data entry was performed using tablets with questionnaires programmed in the ArcGIS Survey123 app. The online version of the same platform was utilized for daily systematization, verification, and correction of inconsistencies. The original anonymized data can be accessed from the S1 Dataset.

### 2.3. Data Analysis

Descriptive analyses were conducted, and the absolute and relative frequency and chi-square values were calculated at the 5% significance level to assess the differences between the proportions of each covariate and the three categories of food security: (1) food security, (2) mild food insecurity, and (3) moderate-to-severe food insecurity. The categories of food security were established in accordance with the Brazilian context during the validation of the scale [39]. Based on this scale, food security was characterized by a household’s access to an adequate and suitable food supply. Mild food insecurity arose when there were concerns about future food availability and the quality of food was impacted. Moderate food insecurity emerged when the diet’s quality was inadequate, and food began to become scarce within the household, often prioritizing the children’s food over the adults’ food. Lastly, severe food insecurity manifested when the quantity of food was insufficient for all occupants, including children. For our analysis, the last category grouped moderate and severe food insecurity because they necessarily represent levels of hunger.

The independent variables elected as correlates with food insecurity were as follows: a wealth index (poorest, intermediate, least poor) that considered housing characteristics and assets, such as vehicles and home appliances, and was computed as described by Filmer and Pritchett [49]; whether the respondent was a beneficiary of a conditional cash transfer program called “Bolsa Família” in Portuguese (no, yes); whether the household self-reported as officially considered impacted by Belo Monte dam construction (no, yes); living in a Collective Urban Resettlement (no, yes); number of household members (1 to 2, 3, 4, 5 or more); if there is any elderly member in the household, i.e., who is 60 years old or over (no, yes); gender of the household head (female, male); skin color of the household head (white, black or brown, other); age of the household head (18–39, 40–59, and 60 years old or over); marital status of the household head (single, divorced or widowed, married or consensual union); education level of the household head (≤4 years, >4 years and ≤11 years, >11 years); and work status of the household head (unemployed, formally employed/employer, informally employed/self-employed/casual worker). 

The selection of correlates was informed by a comprehensive literature review that focused on factors anticipated to be more strongly associated with food insecurity in our study setting. The wealth index served as a representation of the household’s purchasing power and acted as a proxy for the ability to access food in an urbanized area [41,50]. Including beneficiaries of the conditional cash transfer program in our model allowed us to control for this variable, which is a crucial consideration given that the population eligible for this program must exhibit social vulnerability [51,52].

Assessing whether a household has been officially recognized as impacted by the dam helped to determine the extent to which those directly affected by the dam construction may experience food insecurity. Previous studies highlighted that populations directly impacted by dam construction may see a decline in their income production [34,53], potentially leading to repercussions on their food security status.

The Collective Urban Resettlements (Portuguese acronym, Reassentamentos Urbanos Coletivos—RUCs) are neighborhoods created for resettling populations displaced due to the construction of the dam. They can be riverine, rural, or urban populations that had to move due to the increased levels of the Xingu River created by the dam reservoir [32]. Incorporating this information into the model enabled us to pinpoint individuals residing in areas specifically designated for impacted populations. Those residing in resettlements were anticipated to represent socially vulnerable populations [36].

The remaining correlates employed in this study aligned with established parameters in the literature that address food insecurity. These include the number of household members [39,54], the presence of any elderly members in the household [54], the gender of the household head [39,54,55], the skin color of the household head [39,41,50,54,55], the age of the household head [56], the marital status of the household head [55], the education level of the household head [39,55,56], and the work status of the household head [39].

Skin color data were gathered via self-identification following the method employed by the IBGE, which is the institution responsible for conducting Brazilian population censuses. The decision to include this variable in the analysis was driven by its potential use as a proxy for socioeconomic status, income, and educational attainment. Skin color is one element within the nexus of factors that interconnect socioeconomic inequalities and food insecurity in Brazil [41,57,58,59].

The net of associations between the covariates and the main outcome, namely, food insecurity, is shown in Figure 2. Generalized structural equation modeling (GSEM) was used, particularly based on multinomial logistic regressions to allow for understanding of the complex pathways between multiple variables [60]. Variables selection for the final model was based on the hierarchical approach developed by Victora et al. [61], in which we removed the variables that did not reach *p*-value < 0.01 at each level. First, at the distal level, we modeled socioeconomic determinants (as the outcome), considering them as a function of household settings, if the household was officially considered to be impacted by the dam and the head of the household characteristics. We kept only the variables that reached *p*-value < 0.01. Then, we regressed food insecurity (as an outcome), adding to the model the proximal variables of the socioeconomic determinants. We kept from this level only the variables that reached statistical significance. For the wealth index variable, we selected the least poor category as the reference to enhance and broaden the comparison with the poorest stratum. Log-likelihood and the AIC and BIC were utilized as criteria to assess the best model fit, comparing the full model (with all variables) and the one after passing through the variables selection process. Data analyses were conducted using the software Stata v. 15.1 [62].

## 3. Results

The response rate reached 95%, and the 5% refusals were addressed by replacing these households with newly randomly selected ones until achieving a representative sample of 500 households. Table 1 presents the descriptive statistics for questions directly related to food insecurity. Among the 500 households surveyed, 492 yielded valid responses (98.4%), with the remaining households offering incomplete answers. Out of the total valid responses, 48% indicated that in the three months prior to the interview, their residents had concerns about running out of food before being able to purchase or receive more. Additionally, 9% of the respondents reported that in the last three months, at least one adult member of their household had either consumed only one meal a day or had gone a whole day without eating due to a lack of money to buy food. The percentage of positive responses to each item on the psychometric scale followed the theoretical expectation that items related to less severe situations of food insecurity are more frequent than those that show greater difficulty in accessing food. This confirmed the internal consistency (validity) of the answers given to the scale items shown in Table 1. Questions 1 to 8 in Table 1 were sourced from the Brazilian Household Food Insecurity Measurement Scale (EBIA). Consequently, these questions formed the foundation for constructing the food insecurity index employed in this study [48]. 

Regarding the accessibility of the desired amounts and types of food after the conclusion of the Belo Monte dam construction, over 69.7% of the households answered “Yes” to experiencing increased difficulty. Among those who responded positively (*n* = 343), 52.5% acknowledged that this challenge existed even before the onset of the COVID-19 pandemic. In other words, for 36.6% of the total valid household interviews (180 out of 492), food insecurity was already an issue following the dam construction, even preceding the pandemic. 

The evaluation of the food insecurity index demonstrated that more than 28.3% of the households in the Altamira urban area faced moderate or severe food insecurity (Table 2). Mild food insecurity was observed in 32.7% of the households, resulting in a combined total of 61% of households experiencing some level of food insecurity.

As expected, moderate/severe food insecurity was strongly associated with the poorest segment of the wealth index (Table 3). Moreover, certain characteristics of the household head also played a significant role. Female-headed households, those headed by individuals of working age (18–59 years old), single, divorced or widowed heads, heads with lower education levels (≤4 years of study), and unemployed heads were more likely to experience moderate/severe food insecurity. Larger households, particularly those with 5 or more members, were also more vulnerable to food insecurity. Additionally, households that were located in Collective Urban Resettlement neighborhoods, beneficiaries of the cash transfer program, and officially recognized as impacted by the Belo Monte construction were more prone to moderate/severe food insecurity.

Interestingly, households with members aged 60 years or older demonstrated a lower prevalence of food insecurity and the influence of skin color did not exhibit a significant association compared with the categories of the dependent variable.

In the final model, we excluded the variables “marital status of household head” and “skin color of household head”, as they did not reach the significance level of *p* < 0.01 among the distal level variables. Subsequently, when incorporating the proximal level variables, we removed the variable “benefit from cash transfer program”. While the full model, including all available covariates, yielded a log-likelihood of −1312.908, AIC of 2715.802, and BIC of 2904.733, the final model produced a log-likelihood value of −1072.396, AIC of 2212.792, and BIC of 2355.54. These findings indicate that following the variable selection process, the model was more parsimonious and demonstrated an improved fit. The results of the generalized structural equation modeling based on the final model are presented in Table 4.

The findings indicate a strong positive association between moderate or severe food insecurity, as well as mild food insecurity, and the poorest stratum of the population primarily, with a lesser extent to the intermediate stratum. In other words, individuals in the poorest population were the most susceptible to experiencing food insecurity. Analyzing the conditioning factors of these socioeconomic strata, we observed that residents of resettlement areas; households with a larger number of residents (especially 5 or more); households without elderly members; and households headed by females, younger individuals with lower levels of education (≤ 4 years of formal education), and unemployed individuals were more likely to belong to the poorest stratum. Having a household head with more than 11 years of formal education emerged as the strongest protective factor against entering the poorest stratum, even when considering other socioeconomic and demographic variables. The second-strongest protective factor was being formally employed or an employer. Furthermore, the results suggest that households officially considered as impacted by the dam construction had significantly higher chances of being relocated to a resettlement area.

## 4. Discussion

This study sheds light on the complex network of connections associated with food insecurity of the population in the urban area of Altamira, which is located in the Brazilian Amazon. Our findings demonstrate that being officially recognized as impacted by the dam construction significantly increased the likelihood of residing in resettlement areas. Consequently, this amplified the chances of belonging to the most economically disadvantaged social strata, which, in turn, increased the risk of experiencing food insecurity. 

Existing literature supports our initial hypothesis that food insecurity is influenced by a range of underlying factors, which create a layered effect. The attributes of the household head play a crucial role in either exacerbating or mitigating exposure to food insecurity [41]. Published studies have consistently shown that households led by females, individuals of working age, those with lower education levels, and the unemployed are more susceptible to experiencing higher levels of food insecurity [39,41,50,54,55]. Female-headed households bear the weight of women often occupying lower positions in the labor force, leading to disparities in income levels. This, in turn, elevates the social vulnerability of these households, ultimately being associated with higher levels of food insecurity in Brazil [39,41,55], Ethiopia [63], Nigeria [64], South Africa [65], and El Salvador [66], to mention a few examples. Our results corroborate these previous findings.

Interestingly, our research also demonstrated older household heads served as a protective factor against food insecurity when compared with those in the working age bracket, who were more susceptible to market volatilities and unemployment. We also found that the presence of older members (aged 60 years and over) in a household reduces the probability of that household belonging to the poorest stratum of the wealth index. As a result, the presence of older members offered some level of protection against moderate-to-severe food insecurity. These findings can be attributed to the significant contribution of elders’ pensions as an important source of income for households in economically disadvantaged areas. Therefore, older household heads and the presence of an older member can be considered a valuable asset for many underprivileged households, providing a measure of financial stability [67].

Furthermore, supporting our expectations, our study revealed that households with higher crowding, particularly those with five or more members, were more predisposed to experiencing food insecurity [39,54]. As anticipated, recipients of the conditional cash transfer program “Bolsa Família” also exhibited a heightened susceptibility to food insecurity, which was consistent with previous literature [51,52]. This finding was likely attributable to the program’s focus on socially vulnerable populations for enrollment. A complementary explanation is that these populations lived in critical household conditions where social benefits alone could not fully alleviate food insecurity [40].

Moreover, individuals officially recognized as impacted by the Belo Monte construction were more prone to moderate/severe food insecurity. This can be rationalized by previous analyses, which demonstrated that compensation does not necessarily improve the lives of the affected population. On the contrary, dam construction tends to increase population stress, diminish social capital and self-rated health [33,68], and jeopardize livelihoods, including the food supply [34].

Interestingly, marital status and skin color did not attain statistical significance in the model. Previous research indicates that households led by single and black individuals tend to exhibit higher levels of food insecurity [41,55,69]. It is conceivable that the “explanatory power” of these variables was potentially overshadowed in this model by other proxies for socioeconomic status, specifically the wealth index and the condition of receiving benefits from cash transfer programs. Furthermore, a significant majority of respondents fell within the same color group, either black or brown (397 individuals, constituting over 80% of our sample). The limited diversity in responses may have contributed to the consequence that this variable was not able to distinguish population groups within our study area.

Our results supported our second hypothesis that poverty played a pivotal role among the underlying factors, and the most affected groups were those that underwent resettlement. This is in line with prior research that demonstrated a clear association between poverty, which was assessed in our study using a wealth index, and food insecurity [41,50]. We were able to illustrate that the populations residing in resettlement areas faced an elevated likelihood of being in the lowest socioeconomic status, underscoring the detrimental link between resettlement and poverty that exacerbates food insecurity.

Our findings are consistent with national data on food and nutritional insecurity. The II VIGISAN report, which was published in 2022, indicates that families with incomes below half of the minimum wage per capita are the most vulnerable to moderate and severe food insecurity across all Brazilian states. These families often report unemployment or precarious employment and low educational levels of the household head. Additionally, high levels of indebtedness further impede their ability to access food [40].

Interestingly, according to the PENSSAN report in 2022 (the same year when the survey was applied in our study area), the urban areas of Brazil had 57.6% of their households facing mild or moderate/severe food insecurity [40]. Our study showed that this figure was higher in the city of Altamira, amounting to 60%. Considering this, despite progress over recent decades, the country is still heavily unequal in terms of socioeconomic conditions, with large portions of the population living under the poverty line [54,70,71], and the fact that the Brazilian average for food insecurity levels performs better than in Altamira city does not seem to make sense. This is especially true considering the BRL 6.5 billion (~USD 1.3 billion) that dam builders claimed to have invested in the region in “socioenvironmental and sustainability” actions from 2016 to 2022 [72]. The investments either proved insufficient in light of the pre-existing issues in the region—and the ones resulting from the construction—or they were not conducted with due diligence. However, this analysis needs further investigation and falls outside the scope of the current paper.

These results align with previous studies conducted in countries of the Global South, including Lesotho [22], Ethiopia [23], and Laos [24], that also showed that despite the large investments, there is still a disconnection with the human element in dam construction projects following the fact that a significant portion of the local populations affected by the dams remains in a state of food insecurity. Despite promises of improving the quality of life for local populations, the construction of hydropower dams exacerbates displacement and resettlement, resulting in the loss of homes, cultivated and grazing lands, and income and agricultural production. In other words, in areas near dam construction, particularly among resettled populations, livelihoods and food resources undergo abrupt changes, ultimately leading to conditions of food insecurity, which are even more severe for those already experiencing poverty. These findings indicate that the strategies employed by dam builders to address the socio-economic impacts imposed by dam construction on local populations are not delivering the promised outcomes [23,24,73,74,75]. 

Previous research explored the interdependence of water, energy, and food systems, revealing that hydropower can jeopardize food production through several channels. These include competition for the same water resources, a rise in poverty, and the emission of greenhouse gases linked to climate change, resulting in a decline in agricultural production [68,76,77,78,79,80]. It is important to highlight that negative social and environmental issues emerging or increasing in the context of hydropower construction can undermine people’s perception regarding this energy source, which puts the necessary energy transition from more polluting sources to renewables at risk [81,82,83].

Notwithstanding the frequent failures in the monitoring of environmental licensing processes for hydropower dams in the Brazilian Amazon, certain mitigation and compensation actions may have achieved some level of success. Data suggests that malaria control programs, which were implemented as part of the mitigation plan, were strengthened in areas where recent dams were constructed, including the influence area of Belo Monte, resulting in a significant decrease in malaria notifications [84]. Unfortunately, malaria control seems to be an exception, as other public health issues were not given equal priority by dam builders [85].

Another issue is the resettlement process, which relocated riverine populations, where one of their main economic activities was fishing, to the outskirts of Altamira city. The Brazilian Institute of Environment and Renewable Natural Resources (IBAMA) mandated that resettlement areas must be established within a maximum distance of 2 km from the river. However, this requirement was not respected by the dam builders, as they relocated families up to 6 km away from the river without providing convenient and direct transportation to the river shore. Consequently, the resettled populations became “fishers with no river” [34]. This situation has undergone recent changes, which were prompted by a legal mandate that compelled the dam manager, namely, Norte Energia SA, to once again relocate fishermen to new resettlement areas. During our latest field visit in August 2023, we observed that these areas are now situated closer to the river and the occupation by the new residents has started. Indeed, as highlighted in the literature, special attention should be given to food security when addressing resettlement processes resulting from the construction of hydropower dams [86].

Riverine populations in the Amazon heavily rely on fish as a staple food in their daily diet. They typically consume fish six to seven days a week, averaging over 160 kg per capita per year, which is among the highest in the world [5,87]. Nevertheless, this trend is undergoing a shift, which is marked by the substitution of fish consumption with industrially processed food throughout the Amazon region. This emerging pattern not only leads to the erosion of cultural heritage among these communities but also diminishes the nutritional quality of their diets [87]. When riverine populations are displaced and resettled to urban areas far away from the river, they are unable to engage in their traditional economic activity of fishing. As a result, they become reliant on government cash transfers and informal jobs to survive.

In the city, every aspect of life, including food acquisition, becomes monetized [88], challenging the access to food for populations used to fish. Hence, the evidence laid out in this paper implies that the resettled populations were deprived of their fundamental human right to sufficient access to food. Our findings demonstrate that even after factoring in all other sociodemographic variables, living in a resettlement area is linked to being situated within the most disadvantaged socioeconomic category, consequently leading to increased vulnerability to heightened levels of food insecurity.

This study revealed that the construction of the Belo Monte dam actually created one more area where poverty and social deprivation became concentrated. It can be argued that these populations were already living in poverty prior to their relocation. Previously, many of them resided in informal settlements near the city center or riverine areas along the Xingu River [35,89]. Those who lived near the city center had better access to job opportunities, including part-time service jobs. However, after resettlement, being far from the commercial area and facing limited public transportation options, finding work became more challenging. In addition, as previously indicated, the riverine populations resettled in urban areas faced even greater difficulties in reestablishing their economic activities due to being relocated far from the river.

According to the most recent report of the Belo Monte Mitigation Plan (Plano Básico Ambiental—PBA), which is a mandatory and regularly conducted assessment by the dam manager, namely, Norte Energia SA, it was found that during the second semester of 2022, even the remaining fishers of the Xingu River reported a decrease in their fish consumption. Meanwhile, the consumption of milk and processed foods doubled in their daily meals [90]. This change in dietary patterns can be attributed to the decline in fish stocks in the river [5,37,88,91,92,93,94], which has led to an increased reliance on alternative sources of protein and other nutrients. 

A previous study demonstrated the construction of mega dams in another major Amazon River led to a significant reduction in household fish consumption, from a daily basis prior to the dams to only 1–2 times per week or even less frequently after the dams [5].

Our third hypothesis—that post-dam construction, COVID-19 increased the challenges related to food insecurity in the study area—was also corroborated by our findings. During the COVID-19 pandemic, disruptions in food supply chains and the subsequent rise in food prices [95,96,97,98] could have further exacerbated the risk of food insecurity for the populations affected by the dam construction who were striving to rebuild their livelihoods. This was validated by the fact that 69.7% of our sample stated that food access became difficult after the dam construction and, of them, 52.5% mentioned it was already a problem even before the pandemic onset. The finding that over 36.6% of the entire sample acknowledged pre-existing food access issues underscores that more than one-third of the interviewed households linked the dam to their difficulties in accessing food. Another third indicated that COVID-19 undoubtedly worsened their food insecurity status. For the remaining third, food was not an issue. This discovery emphasizes that as stated by residents, the dam could have contributed to the current food insecurity for a parcel of the population, but its impact was significantly intensified by the repercussions of the pandemic. Targeted policies are necessary for the food-insecure population, which constitutes approximately two-thirds of the total population.

In summary, the existing body of literature on dams and resettlement referring to Belo Monte and other dams, consistently points to declining livelihoods, along with other adverse social and environmental effects resulting from dam construction [9,99]. Various studies highlight the strains on community resources and social capital caused by the construction of new dams in the Brazilian Amazon, which have had detrimental implications for self-rated health [32]. These studies also revealed increased groundwater contamination [100]; loss of agricultural labor as the workforce migrates to higher-paying jobs in the dam or associated commercial sector, thereby impacting the food chain and food prices during construction [25]; and the deleterious environmental impacts arising from the rise in greenhouse gas emissions leading to climate change [79].

We contributed to the existing literature by highlighting that food insecurity is a detrimental issue in the context of a hydropower dam post-construction phase for host populations in the Brazilian Amazon, particularly those who were resettled and poor. This aspect has received limited research attention and requires further investigation. 

A potential limitation of our study was that it relied on a cross-sectional analytical approach, which involved data collection at a single point in time. Ideally, it would be desirable to develop a cohort study with follow-ups, utilizing the same food insecurity scale [48,101] applied to the same population. However, comparable data from previous periods are unavailable. Consequently, we cannot make any assumptions regarding how food insecurity levels changed over time. Despite this limitation, our study design and results still sustain the main points reviewed here: (1) even populations that “benefited” from dam construction strategies to reduce its social impacts, such as resettlement, struggle with food insecurity less than 10 years after the dam’s completion, and (2) having 60% of the population facing some level of food insecurity goes in the opposite direction of the promises of “development” made before the dam construction to the local population.

## 5. Conclusions

This study contributed to the existing literature by shedding light on the hidden aspects of food insecurity levels in a locality directly affected by the construction of a large hydropower dam. Our findings reveal that despite the substantial investments made in the largest city affected by the project, namely, Altamira, as required for the installation of the Belo Monte dam, the local population continued to face significant challenges in accessing food. This issue was particularly alarming for those residing in resettlement areas.

Food insecurity can persist or exacerbate following the completion of construction due to a decline in financial resources and the loss of employment opportunities in the local economy [17]. Consequently, while individuals experience further impoverishment, prices do not necessarily revert to their previous levels, which can aggravate the deterioration of local livelihoods.

We can effectively address social issues like food insecurity only when we accurately diagnose them. This study marks a pioneering effort in the literature by employing a scientifically tested scale to evaluate food insecurity within the post-construction phase of a major hydropower dam in the Brazilian Amazon. We propose food security to be recognized as a vital indicator for evaluating the sustainability of large hydropower dams, particularly in the Global South, where these projects are predominantly planned and implemented. We argue that to achieve the sustainable development goals outlined in the 2030 Agenda, it is essential to address not only the apparent social and environmental consequences but also the concealed direct or indirect implications that can undermine food security during and after energy infrastructure construction. 

## Figures and Tables

**Figure 1 ijerph-21-00155-f001:**
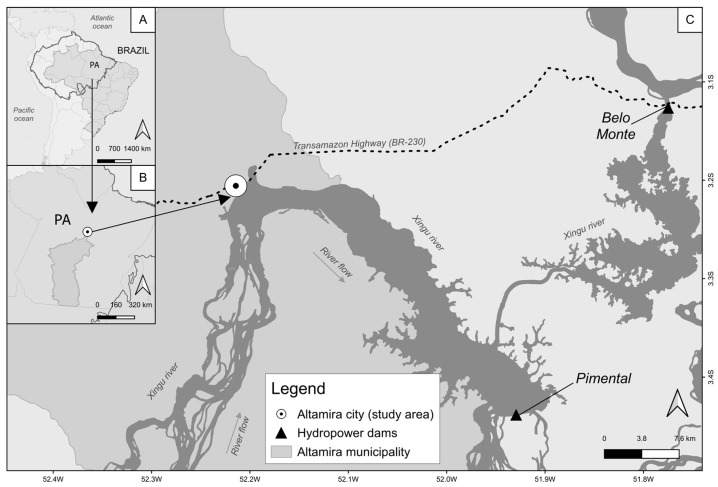
Study area. (**A**) South America, Brazil, and Amazon basin (thick line crossing international borders). (**B**) Pará state, in Brazil, with Altamira municipality highlighted in darker gray. (**C**) Altamira city, the study area, and the close dams Pimental and Belo Monte. Note: Pimental and Belo Monte dams are part of the same hydropower complex. For this reason, in this study, we refer to both as the Belo Monte dam.

**Figure 2 ijerph-21-00155-f002:**
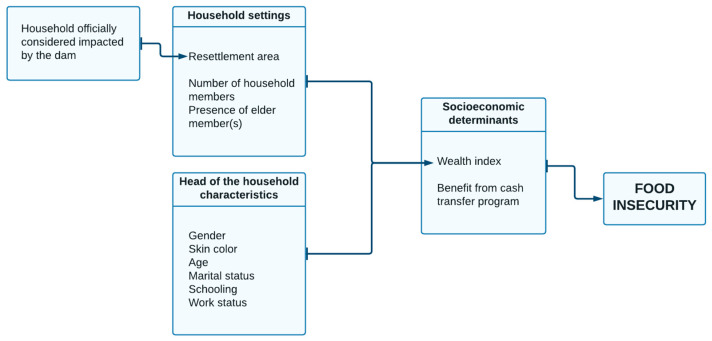
Framework of analysis of food insecurity in Altamira using generalized structural equation modeling. The most proximal correlates of food insecurity are the socioeconomic determinants (household wealth index and whether it received benefit from cash transfer program). The distal covariates used to assess the wealth index were household settings, whether the household was officially considered impacted by the dam, and head of the household characteristics.

**Table 1 ijerph-21-00155-t001:** Summary descriptive statistics on the results of food insecurity in Altamira city, Pará state, Brazil—July 2022.

Questions	No (*n*)	Yes (*n*)	Total (*n*)	No (%)	Yes (%)	Total (%)
(1) In the last three months, have the residents of this household been concerned that food will run out before they can buy or receive more food?	256	236	492	52.0	48.0	100.0
(2) In the last three months, did you run out of food before the residents of this household had the money to buy more food?	328	164	492	66.7	33.3	100.0
(3) In the last three months, did the residents of this household run out of money to have a healthy and varied diet?	275	217	492	55.9	44.1	100.0
(4) In the last three months, did the residents of this household eat only a few types of food that they still had, because the money ran out?	309	183	492	62.8	37.2	100.0
(5) In the last three months, did any resident aged 18 years or older miss a meal because there was no money to buy food?	436	56	492	88.6	11.4	100.0
(6) In the last three months, has any resident aged 18 or over ever eaten less than he/she thought he/she should, because there was no money to buy food?	401	91	492	81.5	18.5	100.0
(7) In the last three months, has any resident aged 18 years or older ever felt hungry but not eaten because there was no money to buy food?	437	55	492	88.8	11.2	100.0
(8) In the last three months, did any resident aged 18 or over ever eat just one meal a day or go a whole day without eating because there was no money to buy food?	448	44	492	91.1	8.9	100.0
(9) Now looking to the past: following the completion of the Belo Monte construction, or since 2016, has access to the amount and types of food that you and your family would like to eat become more difficult?	149	343	492	30.3	**69.7**	100.0
(10) Was this already a problem before the pandemic?	163	180	343	47.5	**52.5**	100.0

**Table 2 ijerph-21-00155-t002:** Scale of food insecurity in the city of Altamira, Pará (Brazil), in the context of post-Belo Monte dam construction—July 2022.

Level of Food Security	Households (*n*)	Households (%)	Number of Times the Household Head Answered “Yes” to the Eight Questions in Food Insecurity Scale *
Food security	192	39.0	0
Mild insecurity	161	32.7	1–3
Moderate/severe insecurity	139	28.3	4–8
Total	492	100.0	

* Note: cut-off points adapted from [48].

**Table 3 ijerph-21-00155-t003:** Descriptive statistics of conditions associated with mild and moderate/severe food insecurity, Altamira (PA), Brazil—July 2022.

Variables	Food Security		Mild Food Insecurity		Moderate/Severe Food Insecurity		Total		*p*-Value
	*n* = 193	39.2%	*n* = 160	32.5%	*n* = 139	28.3%	*n* = 492	100.0%	
*Wealth index*									
Poorest	36	22.2	55	34.0	71	43.8	162	100.0	<0.01
Intermediate	54	32.9	55	33.5	55	33.5	164	100.0	
Least poor	103	62.1	50	30.1	13	7.8	166	100.0	
*Benefit from cash transfer program*									
No	155	47.4	98	30.0	74	22.6	327	100.0	<0.01
Yes	38	23.0	62	37.6	65	39.4	165	100.0	
*Household officially considered impacted by Belo Monte construction*									
No	165	43.3	120	31.5	96	25.2	381	100.0	<0.01
Yes	28	25.2	40	36.0	43	38.7	111	100.0	
*Living in Collective Urban Resettlement*									
No	165	41.7	129	32.6	102	25.8	396	100.0	<0.05
Yes	28	29.2	31	32.3	37	38.5	96	100.0	
*Number of household members*									
1 to 2	83	52.9	36	22.9	38	24.2	157	100.0	<0.01
3	48	38.1	46	36.5	32	25.4	126	100.0	
4	41	35.0	46	39.3	30	25.6	117	100.0	
5 or more	21	22.8	32	34.8	39	42.4	92	100.0	
*Presence of elderly members (60 years old or over)*									
No	114	35.0	120	36.8	92	28.2	326	100.0	<0.01
Yes	79	47.6	40	24.1	47	25.3	166	100.0	
*Gender of household head*									
Female	107	34.1	95	30.3	112	35.7	314	100.0	<0.01
Male	86	48.3	65	36.5	27	15.2	178	100.0	
*Skin color of household head*									
White	34	45.3	28	37.3	13	17.3	75	100.0	0.219
Black or brown	150	37.8	127	32.0	120	30.1	397	100.0	
Other	9	45.0	5	25.0	6	30.0	20	100.0	
*Age of household head*									
18–39 years	44	32.4	53	39.0	39	28.7	136	100.0	<0.05
40–59 years	87	37.7	76	32.9	68	29.4	231	100.0	
60 years or over	62	49.6	31	24.8	32	25.6	125	100.0	
*Marital status of household head*									
Single, divorced, or widowed	73	35.1	64	30.8	71	34.1	208	100.0	<0.05
Married or consensual union	120	42.25	96	33.8	68	23.9	284	100.0	
*Education level of household head*									
≤4 years	48	32.0	43	28.7	59	39.3	150	100.0	<0.01
>4 years and ≤11 years	50	35.0	51	35.7	42	29.4	143	100.0	
>11 years	95	47.7	66	33.2	38	19.1	199	100.0	
*Work status of household head*									
Unemployed	78	36.5	62	29.0	74	34.6	214	100.0	<0.01
Formally employed/employer	53	54.6	27	27.8	17	17.5	97	100.0	
Informally employed/self-employed/casual worker	62	34.3	71	39.2	48	26.5	181	100.0	

**Table 4 ijerph-21-00155-t004:** Generalized structural equation modeling analysis of correlates of food insecurity in Altamira, Brazil—July 2022 (*n* = 492).

			Coef.	*p*-Value	95% Conf. Interval
Mild insecurity (ref. food security)	Wealth index (ref. least poor)	Poorest	1.15	<0.01	0.61–1.69
		Intermediate	0.74	<0.01	0.24–1.25
Moderate and severe food insecurity (ref. food security)	Wealth index (ref. least poor)	Poorest	2.75	<0.01	2.05–3.45
		Intermediate	2.09	<0.01	1.40–2.78
Wealth index—poorest (ref. least poor)	Resettlement area (ref. no)	Yes	1.78	<0.01	0.79–2.77
	Number of household members (ref. 1–2)	3	1.47	<0.01	0.67–2.23
		4	1.74	<0.01	0.92–2.56
		5 or more	2.63	<0.01	1.69–3.59
	Presence of elder members (ref. no)	Yes	−1.20	<0.05	−2.67–−0.12
	Gender of the head of the household (ref. female)	Male	−0.80	<0.05	−1.44–−0.19
	Age of the head of the household (ref. 18–39)	40-59	−0.88	<0.05	−1.56–−0.19
		60 or over	−1.61	<0.05	−2.95–−0.28
	Education level of the head of the household (ref. ≤ 4 years)	>4 and ≤11	−0.80	0.05	−1.62–−0.02
		>11	−2.95	<0.01	−3.79–−2.09
	Work status of the head of the household (ref. unemployed)	Formally employed/employer	−1.82	<0.01	−2.72–−0.90
		Informally employed/ self-employed/casual worker	−0.23	0.50	−0.90–0.44
Wealth index—intermediate (ref. least poor)	Resettlement area (ref. no)	Yes	1.87	<0.01	0.93–2.80
	Number of household members (ref. 1–2)	3	−0.05	0.87	−0.67–0.56
		4	0.06	0.85	−0.59–0.72
		5 or more	0.21	0.61	−0.63–1.06
	Presence of elder members (ref. no)	Yes	0.01	0.98	−0.90–0.92
	Gender of the head of the household (ref. female)	Male	−0.52	0.05	−1.03–−0.01
	Age of the head of the household (ref. 18–39)	40–59	−0.60	0.05	−1.22–0.02
		60 or over	−1.28	<0.05	−2.42–−0.15
	Education level of the head of the household (ref. ≤ 4 years)	>4 and ≤11	−0.65	0.08	−1.39–0.09
		>11	−1.93	<0.01	−2.66–−1.21
	Work status of the head of the household (ref. Unemployed)	Formally employed/employer	−1.61	0.11	−1.27–0.13
		Informally employed/ self-employed/casual worker	0.22	0.47	−0.38–0.85
Resettlement area (ref. no)	Household officially considered impacted (ref. no)	Yes	3.23	<0.01	2.67–3.79

## Data Availability

Data containing the original anonymized data used in our analyses are available in the S1 dataset.

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
