# Peer review of "Poverty–Food Insecurity Nexus in the Post-Construction Context of a Large Hydropower Dam in the Brazilian Amazon"

_ijerph, 2024, doi:10.3390/ijerph21020155_

Round 1

Reviewer 1 Report

Comments and Suggestions for Authors

This study evaluate post-construction food security situation in the area directly affected by the construction of Brazil's second-largest dam, Belo Monte (2011-2016), in the Amazon's Pará state. The article herein deals with an interesting and timely topic. Yet, in its current version, the paper has got minor issues that need to be addressed before it goes through the next phase. I elaborate on my concerns below and hope that they help authors strengthen their contributions.

1.     The weakest point of the paper is theoretical background of the paper. It could be nice to add a brief info about how the variables have been chosen. It would have been more appropriate for the authors to evaluate their hypotheses according to theory and similar studies in the literature.

2.     Is the classification of education levels such as “low” (less than or equal to 4 years of schooling) based on authors’ discretion or already existing and widely regarded standard(s)? [296-298]

3.      There exist some findings that are not adequately touched on or elaborated on in the discussion section. For example, why female-headed households are more susceptible to increased risk of food insecurity in comparison to male-headed ones? The authors nicely elaborated on the finding associated with households with elderly aged 60 or more being less susceptible to food insecurity [466-473]. In the same manner, findings associated with each household characteristic need to be elaborated on for potential readers. [296-312]

4.     One of the socio-economic determinants, the wealth index, is rather abstract. I have observed 4 categories whose specifications I do not comprehend; namely, richest, intermediate, least poor, and poorest. What measure do authors use to categorize the wealth index into the mentioned four? What is the difference between the poor and the least poor? [220-222] & [296-298].

5.     The authors’ decision to incorporate skin color is due to “its potential use as a proxy for socioeconomic status, income, and educational attainment”, as justified.  How is skin color associated with income, status, and educational attainment? What is the theoretical justification given that skin color differences are not statistically associated with food insecurity? [296-298]&[224-231]

Author Response

We sincerely appreciate the constructive comments and suggestions provided by the reviewer. The reviewer’s comments are highlighted in blue and our answers to each comment are in black font. We have revised the manuscript in order to address all concerns raised and here we provide detailed responses to all comments and suggestions. 

Questions for General Evaluation Reviewer’s Evaluation Response and Revisions
  Yes Can be improved Must be improved Not applicable  
Does the introduction provide sufficient background and include all relevant references? ( ) (x) ( ) ( ) We have enhanced the introduction by incorporating additional background information and references.
Are all the cited references relevant to the research? (x) ( ) ( ) ( ) Thank you.
Is the research design appropriate? ( ) (x) ( ) ( ) We have included details about our study setting and design to enhance clarity.
Are the methods adequately described? ( ) (x) ( ) ( ) We refined the description of the methods, focusing specifically on the aspects of data collection and data analysis.
Are the results clearly presented? ( ) (x) ( ) ( ) We have thoroughly reviewed the results to enhance clarity.
Are the conclusions supported by the results? (x) ( ) ( ) ( ) Thank you.

Reviewers' Comments:

This study evaluate post-construction food security situation in the area directly affected by the construction of Brazil's second-largest dam, Belo Monte (2011-2016), in the Amazon's Pará state. The article herein deals with an interesting and timely topic. Yet, in its current version, the paper has got minor issues that need to be addressed before it goes through the next phase. I elaborate on my concerns below and hope that they help authors strengthen their contributions.

Thank you sincerely for dedicating your valuable time and effort to offering insights for the enhancement of our paper.

1. The weakest point of the paper is theoretical background of the paper. It could be nice to add a brief info about how the variables have been chosen. It would have been more appropriate for the authors to evaluate their hypotheses according to theory and similar studies in the literature.

Thank you for your recommendation. We have enhanced the paper by providing additional details to ensure a closer alignment between the selected variables and the theoretical framework. This refinement is particularly evident in the Data Analysis section, where we elucidate the rationale behind selecting specific correlates. Additionally, we cite relevant literature that informed our decision to incorporate these variables, as demonstrated below:

The selection of correlates was informed by a comprehensive literature review, focusing on factors anticipated to be more strongly associated with food insecurity in our study setting. The wealth index serves as a representation of the household's purchasing power, acting as a proxy for the ability to access food in an urbanized area [41,50]. Including beneficiaries of the conditional cash transfer program in our model allows us to control for this variable, a crucial consideration given that the population eligible for this program must exhibit social vulnerability [51,52].

Assessing whether a household has been officially recognized as impacted by the dam helps determine the extent to which those directly affected by the dam construction may experience food insecurity. Previous studies have highlighted that populations directly impacted by dam construction may see a decline in their income production [34,53], potentially leading to repercussions on their food security status.

The Collective Urban Resettlements (Portuguese acronym, Reassentamentos Urbanos Coletivos - RUCs) are neighborhoods created for resettling populations displaced due to the construction of the dam. They can be riverine, rural, or urban populations that had to move due to the increased levels of the Xingu River created by the dam reservoir [32]. Incorporating this information into the model enables us to pinpoint individuals residing in areas specifically designated for impacted populations. Those residing in resettlements are anticipated to represent socially vulnerable populations [36].

The remaining correlates employed in this study align with established parameters in the literature addressing food insecurity. These include the number of household members [39,54], the presence of any elderly members in the household [54], the gender of the household head [39,54,55], the skin color of the household head [39,41,50,54,55], the age of the household head [56], the marital status of the household head [55], the education level of the household head [39,55,56], and the work status of the household head [39].

In addition, we also added information to the Discussion section, improving the evaluation of our hypotheses according to the literature review, as follows:

Existing literature supports our initial hypothesis that food insecurity is influenced by a range of underlying factors, which create a layered effect. The attributes of the household head play a crucial role in either exacerbating or mitigating exposure to food insecurity [41]. Published studies have consistently shown that households led by females, individuals of working age, those with lower education levels, and the unemployed are more susceptible to experiencing higher levels of food insecurity [39,41,50,54,55]. Female-headed households bear the weight of women often occupying lower positions in the labor force, leading to disparities in income levels. This, in turn, elevates the social vulnerability of these households, ultimately being associated with higher levels of food insecurity in Brazil [39,41,55], Ethiopia [63], Nigeria [64], South Africa [65], and El Salvador [66], to mention a few examples. Our results corroborate these previous findings.

Interestingly, our research has also demonstrated older household heads serve as a protective factor against food insecurity when compared to those in the working age bracket, who are more susceptible to market volatilities and unemployment. We also found that the presence of older members (aged 60 years and over) in a household reduces the probability of that household belonging to the poorest stratum of the wealth index. As a result, the presence of older members offers some level of protection against moderate to severe food insecurity. These findings can be attributed to the significant contribution of elders' pensions as an important source of income for households in economically disadvantaged areas. Therefore, older household heads and the presence of an older member can be considered as a valuable asset for many underprivileged households, providing a measure of financial stability [67].

Furthermore, supporting our expectations, our study revealed that households with higher crowding, particularly those with 5 or more members, are more predisposed to experiencing food insecurity [39,54]. As anticipated, recipients of the conditional cash transfer program “Bolsa Família” also exhibit a heightened susceptibility to food insecurity, consistent with previous literature [51,52]. This finding is likely attributable to the program's focus on socially vulnerable populations for enrollment. A complementary explanation is that these populations live in critical household conditions where social benefits alone cannot fully alleviate food insecurity [40].

Moreover, individuals officially recognized as impacted by the Belo Monte construction are more prone to moderate/severe food insecurity. This can be rationalized by previous analyses, which demonstrated that compensation does not necessarily improve the lives of the affected population. On the contrary, dam construction tends to increase population stress, diminish social capital and self-rated health [33,68], and jeopardize livelihoods, including the food supply [34].

Interestingly, marital status and skin color did not attain statistical significance in the model. Previous research has indicated that households led by single and black individuals tend to exhibit higher levels of food insecurity [41,55,69]. It is conceivable that the "explanatory power" of these variables was potentially overshadowed in the model by other proxies for socioeconomic status, specifically the wealth index and the condition of receiving benefits from cash transfer programs. Furthermore, a significant majority of respondents fell within the same color group, either black or brown (397 individuals, constituting over 80% of our sample). The limited diversity in responses may have contributed to the consequence that this variable was not able to distinguish population groups within our study area.

Our results support our second hypothesis that poverty plays a pivotal role among the underlying factors, and the strongest affected groups are those that undergo resettlement. It is in line with prior research that demonstrated a clear association between poverty, assessed in our study through a wealth index, and food insecurity [41,50]. We were able to illustrate that populations residing in resettlement areas faced an elevated likelihood of being in the lowest socioeconomic status, underscoring the detrimental link between resettlement and poverty that exacerbates food insecurity.

Our findings are consistent with national data on food and nutritional insecurity. The II VIGISAN report, published in 2022, indicates that families with incomes below half of the minimum wage per capita are the most vulnerable to moderate and severe food insecurity across all Brazilian states. These families often report unemployment or precarious employment and low educational levels of the household head. Additionally, high levels of indebtedness further impede their ability to access food [40].

Interestingly, according to the PENSSAN report in 2022 (the same year when the survey was applied in our study area), the urban areas of Brazil presented 57.6% of their households facing mild or moderate/severe food insecurity [40]. Our study has shown that this figure was higher in the city of Altamira, amounting 60%. Considering that, despite progress over the last decades, the country is still heavily unequal in terms of socioeconomic conditions, with large portions with populations living under the poverty line [54,70,71], having the Brazilian average of food insecurity levels performing better than the Altamira city does not seem to make sense. This is true especially considering the R$ 6.5 billion (~ $1.3 billion) that dam builders claim to have invested in the region in “socioenvironmental and sustainability” actions from 2016 to 2022 [72]. The investments either proved insufficient in light of the pre-existing issues in the region – and the ones resulting from the construction –, or they were not conducted with due diligence. However, this analysis needs further investigation and falls outside the scope of the current paper.

[…]

Our third hypothesis – that post-dam construction, COVID-19 increased the challenges related to food insecurity in the study area – was also corroborated by our findings. During the COVID-19 pandemic, disruptions in food supply chains and the subsequent rise in food prices [96–100] could have further exacerbated the risk of food insecurity for the populations affected by the dam construction who were striving to rebuild their livelihoods. This is validated by the fact that 69.7% of our sample referred that food access became difficult after the dam construction and, of them, 52.5% mentioned it was already a problem even before the pandemic onset. The finding that over 36.6% of the entire sample acknowledges pre-existing food access issues underscores that more than one-third of the interviewed households link the dam to their difficulties in accessing food. Another third indicated that COVID-19 undoubtedly worsened their food insecurity status. For the remaining one third food was not an issue. This discovery emphasizes that, as stated by residents, the dam could have contributed to the current food insecurity for a parcel of the population, but its impact was significantly intensified by the repercussions of the pandemic. Targeted policies are necessary for the food-insecure population, constituting approximately two-thirds of the total population.

2. Is the classification of education levels such as “low” (less than or equal to 4 years of schooling) based on authors’ discretion or already existing and widely regarded standard(s)? [296-298]

Thank you for your question. Our categorization process was guided by the organizational structure of the education system in Brazil, which comprises three main levels: fundamental, high school, and technical/university. The initial segment of fundamental school spans up to the fourth grade, leading us to group individuals with less than four years of schooling, including those who are entirely illiterate, under the category of the lowest education level.

High school education in Brazil spans a total of 12 years of schooling. Consequently, we defined the intermediate education level as individuals with more than four years but less than or equal to 11 years of schooling. This encompasses those with some educational background who did not complete high school. Individuals with more than 11 years of schooling include those who successfully completed high school and were combined with populations capable of pursuing higher education, representing the most educated segment of the population.

It is worth noting that different studies may employ varying schooling categories. An alternative approach, as seen in one of the studies referenced in our paper (https://doi.org/10.1371/journal.pgph.0002324), involves categorizing education levels as illiterate, 1–7 years, 8–12 years, and >12 years. Utilizing these categories would yield the following results:

a) illiterate = 10.2%

b) 1–7 years = 33.1%

c) 8–12 years = 41.1%

d) >12 years = 15.6%

Alternatively, the selected classes in our study resulted in three distinct categories, dividing the sample as follows:

a) <=4 years = 30.5%

b) >4 years and <=11 years = 29.1%

c) >11 years = 40.4%

We modified the schooling classification to ensure a more balanced distribution of populations within each class. This adjustment was necessary as the illiterate individuals and those with university/technical degrees were largely equivalent. Consequently, we amalgamated them into other groups, enhancing the explanatory power of this variable while reducing the number of categories. In essence, our choice of classification for this variable was informed by both the existing literature and the specific characteristics of the study location and dataset. This decision maintains consistency by creating meaningful and easily explainable classes that underscore the significance of this variable in the study.

3. There exist some findings that are not adequately touched on or elaborated on in the discussion section. For example, why female-headed households are more susceptible to increased risk of food insecurity in comparison to male-headed ones? The authors nicely elaborated on the finding associated with households with elderly aged 60 or more being less susceptible to food insecurity [466-473]. In the same manner, findings associated with each household characteristic need to be elaborated on for potential readers. [296-312]

Thank you for this question, which provides an opportunity to enhance the discussion in our paper. A growing body of literature suggests that female heads of households often find themselves in socially vulnerable conditions, even worse if they are black (please refer to reference II VIGISAN: Suplemento II, Insegurança Alimentar e desigualdades de raça/cor da pele e gênero, 2023). Typically, these individuals are single women facing the challenge of sustaining their families through precarious jobs and/or social benefits, all while managing household responsibilities and caring for their children. While the specific context may vary, the association between social vulnerability and female-headed households has been observed in different parts of the world. We have improved our discussion as follows:

Existing literature supports our initial hypothesis that food insecurity is influenced by a range of underlying factors, which create a layered effect. The attributes of the household head play a crucial role in either exacerbating or mitigating exposure to food insecurity [41]. Published studies have consistently shown that households led by females, individuals of working age, those with lower education levels, and the unemployed are more susceptible to experiencing higher levels of food insecurity [39,41,50,54,55]. Female-headed households bear the weight of women often occupying lower positions in the labor force, leading to disparities in income levels. This, in turn, elevates the social vulnerability of these households, ultimately being associated with higher levels of food insecurity in Brazil [39,41,55], Ethiopia [63], Nigeria [64], South Africa [65], and El Salvador [66], to mention a few examples. Our results corroborate these previous findings.

Interestingly, our research has also demonstrated older household heads serve as a protective factor against food insecurity when compared to those in the working age bracket, who are more susceptible to market volatilities and unemployment. We also found that the presence of older members (aged 60 years and over) in a household reduces the probability of that household belonging to the poorest stratum of the wealth index. As a result, the presence of older members offers some level of protection against moderate to severe food insecurity. These findings can be attributed to the significant contribution of elders' pensions as an important source of income for households in economically disadvantaged areas. Therefore, older household heads and the presence of an older member can be considered as a valuable asset for many underprivileged households, providing a measure of financial stability [67].

Furthermore, supporting our expectations, our study revealed that households with higher crowding, particularly those with 5 or more members, are more predisposed to experiencing food insecurity [39,54]. As anticipated, recipients of the conditional cash transfer program “Bolsa Família” also exhibit a heightened susceptibility to food insecurity, consistent with previous literature [51,52]. This finding is likely attributable to the program's focus on socially vulnerable populations for enrollment. A complementary explanation is that these populations live in critical household conditions where social benefits alone cannot fully alleviate food insecurity [40].

4. One of the socio-economic determinants, the wealth index, is rather abstract. I have observed 4 categories whose specifications I do not comprehend; namely, richest, intermediate, least poor, and poorest. What measure do authors use to categorize the wealth index into the mentioned four? What is the difference between the poor and the least poor? [220-222] & [296-298].

We apologize for any confusion. The wealth index comprises three categories: Poorest, Intermediate, and Least Poor (refer to Table 3). There is no category labeled "Richest." The confusion arose due to our use of "Richest" in Table 4 as a synonym for the highest socioeconomic status, namely, "Least Poor." To eliminate any ambiguity, we have replaced "Richest" with "Least Poor" in Table 4.

5. The authors’ decision to incorporate skin color is due to “its potential use as a proxy for socioeconomic status, income, and educational attainment”, as justified. How is skin color associated with income, status, and educational attainment? What is the theoretical justification given that skin color differences are not statistically associated with food insecurity? [296-298]&[224-231]

Thank you for your question. We have included theoretical background to justify the incorporation of skin color into our study, as seen below in the "data analysis" section.

The remaining correlates employed in this study align with established parameters in the literature addressing food insecurity. These include the number of household members [39,54], the presence of any elderly members in the household [54], the gender of the household head [39,54,55], the skin color of the household head [39,41,50,54,55], the age of the household head [56], the marital status of the household head [55], the education level of the household head [39,55,56], and the work status of the household head [39]. 

Skin color data was gathered via self-identification, following the method employed by IBGE, the institution responsible for conducting Brazilian population censuses. The decision to include this variable in the analysis was driven by its potential use as a proxy for socioeconomic status, income, and educational attainment. Skin color is one element within the nexus of factors that interconnect socioeconomic inequalities and food insecurity in Brazil [41,57–59].

And the text incorporated into the discussion section:

Interestingly, marital status and skin color did not attain statistical significance in the model. Previous research has indicated that households led by single and black individuals tend to exhibit higher levels of food insecurity [41,55,69]. It is conceivable that the "explanatory power" of these variables was potentially overshadowed in the model by other proxies for socioeconomic status, specifically the wealth index and the condition of receiving benefits from cash transfer programs. Furthermore, a significant majority of respondents fell within the same color group, either black or brown (397 individuals, constituting over 80% of our sample). The limited diversity in responses may have contributed to the consequence that this variable was not able to distinguish population groups within our study area.

Thank you once again for your review. We firmly believe that your numerous contributions have significantly enhanced the quality of our paper.

Reviewer 2 Report

Comments and Suggestions for Authors

The manuscript evaluates the impacts of hydropower dams on food security in Brazil. The following recommendations will improve the quality of the manuscript.

1) Authors need to add a detailed description of the hydropower dam showing relevant characteristics such as storage capacity, Design Capacity, Number of turbines and spill gates etc. 

2) It would be good if we could get river flow data below the dam to show how dam construction influences river flow:

3)Authors need to create a concise table that presents the 8 major variables of the questionnaire

Author Response

We sincerely appreciate the constructive comments and suggestions provided by the reviewer. The reviewer’s comments are highlighted in blue and our answers to each comment are in black font. We have revised the manuscript in order to address all concerns raised and here we provide detailed responses to all comments and suggestions.

Questions for General Evaluation

Reviewer’s Evaluation

Response and Revisions

Yes

Can be improved

Must be improved

Not applicable

Does the introduction provide sufficient background and include all relevant references?

(x)

( )

( )

( )

Thank you.

Are all the cited references relevant to the research?

(x)

( )

( )

( )

Thank you.

Is the research design appropriate?

(x)

( )

( )

( )

Thank you.

Are the methods adequately described?

(x)

( )

( )

( )

Thank you.

Are the results clearly presented?

(x)

( )

( )

( )

Thank you.

Are the conclusions supported by the results?

(x)

( )

( )

( )

Thank you.

Reviewers' Comments:

The manuscript evaluates the impacts of hydropower dams on food security in Brazil. The following recommendations will improve the quality of the manuscript.

Thank you very much for providing this review.

1) Authors need to add a detailed description of the hydropower dam showing relevant characteristics such as storage capacity, Design Capacity, Number of turbines and spill gates etc.

We added to the text additional information on the Belo Monte hydropower dam, as follows:

As the second largest hydropower dam in Brazil, Belo Monte boasts an installed capacity of 11.2 GW, generating an average of 4.5 GW. Its structure is equipped with 24 turbines and 18 spillway gates. Designed as a run-of-river hydropower system, it aims to minimize the size of its water reservoir. Nevertheless, the project's flooded area spans a substantial 478 square kilometers [44].

2) It would be good if we could get river flow data below the dam to show how dam construction influences river flow:

Thank you for this recommendation. We added the following information to the text:

The construction had significant social-ecological impacts, notably a substantial reduction in water flow in a specific stretch of about 100 km of the river known as Volta Grande do Xingu. In this area, the river flow decreased by approximately 80% due to the redirection of the natural water flow to the diversion channel leading to the main powerhouse [34,45,46].

3)Authors need to create a concise table that presents the 8 major variables of the questionnaire

Thank you for your suggestion. It's possible that, for some reason, it may not have appeared in your version of the paper. However, Table 1 already displays the eight primary variables from the questionnaire, serving as the foundation for constructing the food insecurity index. To enhance its accessibility in the paper, we have included the following introductory text for Table 1: “Questions 1 to 8 in Table 1 were sourced from the Brazilian Household Food Insecurity Measurement Scale (EBIA). Consequently, these questions formed the foundation for constructing the food insecurity index employed in this study.”

Thank you once again for taking the time to review our paper. We believe that your suggestions have enhanced the overall quality of our manuscript.

Reviewer 3 Report

Comments and Suggestions for Authors

1. Please make consistent objective both in abstract and introduction.

2. The stories and previous studies about a large hydropower dam in the Brazilian Amazon should be improved

3.  This study is panel data from 2016-2022?

4. Why did this study hypothesize food insecurity not food security?

5. If focus on food insecurity, why did not use Food Insecurity Experience Scale Analysis? 

6. Please show the p-value and other statistics test at SEM analysis result

Author Response

We sincerely appreciate the constructive comments and suggestions provided by the reviewer. The reviewer’s comments are highlighted in blue and our answers to each comment are in black font. We have revised the manuscript in order to address all concerns raised and here we provide detailed responses to all comments and suggestions.

Questions for General Evaluation

Reviewer’s Evaluation

Response and Revisions

Yes

Can be improved

Must be improved

Not applicable

Does the introduction provide sufficient background and include all relevant references?

( )

( )

(x)

( )

We have enhanced the introduction by incorporating additional background information and references.

Are all the cited references relevant to the research?

( )

( )

(x)

( )

We have eliminated certain references and incorporated additional ones as per the reviewers' requests.

Is the research design appropriate?

( )

(x)

( )

( )

We have included details about our study setting and design to enhance clarity.

Are the methods adequately described?

( )

( )

(x)

( )

We refined the description of the methods, focusing specifically on the aspects of data collection and data analysis.

Are the results clearly presented?

( )

( )

(x)

( )

We have  reviewed the results to enhance clarity.

Are the conclusions supported by the results?

( )

( )

(x)

( )

We have refined the conclusions to align them with the results and the overarching discussion of the paper.

Reviewers' Comments:

1. Please make consistent objective both in abstract and introduction.

We really appreciate your time and efforts for providing this review. Thank you very much for that. We have changed the text to improve consistency between objective in the abstract and the introduction.

The abstract:

Our study focuses on food insecurity and evaluates this issue in the city of Altamira, in the Brazilian Amazon, which has been profoundly impacted socially and economically by the construction of Brazil's second-largest dam, Belo Monte (2011-2016).

The Introduction:

Belo Monte dam was built between 2011 and 2016. This paper aims to evaluate the food insecurity situation in the city of Altamira, located in the Brazilian Amazon, which has been profoundly impacted socially and economically by the construction of this dam.

2. The stories and previous studies about a large hydropower dam in the Brazilian Amazon should be improved

Thank you for your suggestion. We have incorporated the following paragraph into our Introduction. While these papers were predominantly cited in the discussion, unfortunately, they were inadvertently omitted from the initial section of our paper—an aspect crucial to the overall coherence of our work.

Among these issues, previous studies have revealed that hydropower dams constructed in the Amazon region have resulted in increased stress among local populations [31]. Furthermore, they have been associated with the erosion of social capital [32] and a decrease in self-rated health [33]. Through resettlement, these dams have been found to undermine the cultural, social, and economic reproduction of traditional peoples and communities [34,35]. Additionally, they contribute to gender imbalances [36,37] and impact the food security of local populations [30].

3. This study is panel data from 2016-2022?

Rather than a panel study, this paper utilizes cross-sectional data, collected in July 2022. We moved up the following text to emphasize the study design: “Data collection occurred between July 13th and July 30th, 2022, and was carried out by eight interviewers who were undergraduate or graduate students from local universities. Prior to the household visits, interviewers underwent intensive training.”

4. Why did this study hypothesize food insecurity not food security?

We hypothesize that food insecurity may persist, as it has been demonstrated as an unintended consequence of hydropower projects worldwide, as discussed in our Discussion section:

These results align with previous studies conducted in countries of the Global South, including Lesotho [22], Ethiopia [23], and Laos [24], also showing that despite the large investments, there is still a disconnection with the human element in dam construction projects following the fact that a significant portion of the local populations affected by the dams remains in a state of food insecurity.

5. If focus on food insecurity, why did not use Food Insecurity Experience Scale Analysis?

Certainly, the FAO has recently undertaken a global standardization effort for an eight-item scale known as the Food Insecurity Experience Scale (FIES). This instrument was derived from the eight adult items found in the Latin American and Caribbean Food Security Scale, which itself was significantly influenced by EBIA. FIES has gained recognition as the official food security scale for monitoring one of the Sustainable Development Goals targets and has been implemented in more than 150 countries. Nevertheless, upon scrutinizing the FIES items and the EBIA items, it becomes evident that there is significant overlap, as shown below.

FIES:

During the last 12 months, was there a time when, because of lack of money or other resources:

1- You were worried you would not have enough food to eat?

2- You were unable to eat healthy and nutritious food?

3- You ate only a few kinds of foods?

4- You had to skip a meal?

5- You ate less than you thought you should?

6- Your household ran out of food?

7- You were hungry but did not eat?

8- You went without eating for a whole day?

EBIA:

1- In the last three months, have the residents of this household been concerned that food will run out before they can buy or receive more food?

2- In the last three months, did you run out of food before the residents of this household had the money to buy more food?

3- In the last three months, did the residents of this household run out of money to have a healthy and varied diet?

4- In the last three months, did the residents of this household eat only a few types of food that they still had, because the money ran out?

5- In the last three months, did any resident aged 18 years or older miss a meal because there was no money to buy food?

6- In the last three months, has any resident aged 18 or over ever eaten less than he/she thought he/she should, because there was no money to buy food?

7- In the last three months, has any resident aged 18 years or older ever felt hungry but not eaten because there was no money to buy food?

8- In the last three months, did any resident aged 18 or over ever eat just one meal a day or go a whole day without eating because there was no money to buy food?

Therefore, in terms of results, the choice between applying either scale is not expected to alter the overall outcome. Additionally, we opted for EBIA due to its consistent application across the country for over two decades, continuously refined to better reflect the Brazilian context in evaluating the food insecurity status of the population.

6. Please show the p-value and other statistics test at SEM analysis result

I regret to note that, for some reason, part of the tables (or their entire content) may not have been visible for your evaluation. Table 3, containing descriptive statistics, includes the p-values of the chi-square tests. Furthermore, in Table 4, featuring SEM results, p-values are provided between columns containing coefficients and their corresponding 95% confidence intervals.

Once again, we sincerely appreciate the time you have dedicated to reviewing our paper. With your valuable input, we are confident that the manuscript is now more consistent and well-suited for publication, upon your approval.